# The Immediate Effects of Instrument-Assisted Soft Tissue Mobilization on Pain and Function in Female Runners with Patellofemoral Pain

**DOI:** 10.3390/medicina61111912

**Published:** 2025-10-24

**Authors:** Seong Chan Cho, Young Kyun Kim

**Affiliations:** Graduate School of Sports Medicine, CHA University, Seongnam 13496, Republic of Korea; whtjdcks0104@naver.com

**Keywords:** patellofemoral pain, IASTM, immediate effects, female runners

## Abstract

*Background and Objectives*: Patellofemoral pain (PFP) is the most prevalent running-related injury due to underlying biomechanical factors, particularly among female runners. Although instrument-assisted soft tissue mobilization (IASTM) is a popular therapeutic technique, the optimal application site for the short-and long-term outcomes of PFP has not been well established. This aim of this study was to compare the immediate and short-term (1-week) effects of a single IASTM treatment applied to the hip and knee versus the knee alone on running-related pain. Range of motion (ROM), muscle strength, and functional performance were also assessed to compare change between the two treatment conditions. *Materials and Methods*: Twenty-eight female runners with PFP were randomly assigned to either the Hip and Knee (HK) group (n = 14) or the knee-only (K) group (n = 14). The HK group received a 7-min IASTM treatment targeting the quadriceps, patella, iliotibial band (ITB), and gluteus medius, whereas the K group received a 3-min treatment targeting the quadriceps and patella. Visual analog scale (VAS), hip adduction ROM, hip abduction/external rotation strength, and step-down test scores were measured at baseline, immediately post-intervention, and 1 week later. *Results*: Running-related pain significantly decreased in both groups (main effect of time, *p* < 0.001) from baseline (HK: 5.49 ± 2.14 [95% CI: 4.78–6.68]; K: 5.30 ± 1.45 [95% CI: 4.69–5.91]) to week 1 (HK: 1.30 ± 1.08 [95%CI: 0.69–1.90]; K: 1.57 ± 1.20 [95%CI: 0.93–2.21]). However, no significant difference was found between the groups. Significant improvement was also observed in hip adduction ROM (*p* < 0.001), hip abduction strength (*p* = 0.02), step-down pain (*p* < 0.001), and patellofemoral function (*p* < 0.001) immediately after the intervention, which was sustained at the 1-week follow-up. However, no significant difference was found between the groups. Also, hip external rotation strength showed no significant change over time or between groups (*p* = 0.737). *Conclusions*: A single IASTM session effectively reduced pain and improved function in female runners with PFP. However, the hip treatment did not show a significant additional benefit compared with knee treatment alone. IASTM can provide immediate and short-term relief of pain and functional limitations.

## 1. Introduction

Running is a popular physical activity with a high risk of overuse injuries [1]. This repetitive nature of this sport makes runners susceptible to chronic overuse injuries as the continuous load on fascial structures leads to an accumulation of tension within the body [1,2]. These conditions, particularly in the knee and feet, are very common. Knee injuries account for up to 50% of all running-related injuries, and patellofemoral pain (PFP) is the most prevalent (45.3%) of all knee injuries [2,3,4]. Among non-ultramarathon runners, PFP has the highest prevalence at 16.7% of all running-related injuries [5]. Furthermore, due to lower extremity biomechanical differences, the prevalence of PFP is higher in females, regardless of age [6]. The factors contributing to PFP in females include femoral neck anteversion, genu valgum, genu recurvatum, leg length discrepancy, and increased Q-angle [3,7,8]. In particular, female runners exhibit greater dynamic knee valgus than male runners during running, and this may play a critical role in the development of PFP [8,9]

Furthermore, excessive dynamic knee valgus is often observed owing to femoral internal rotation, hip adduction, and tibial external rotation during running [8,10]. These kinematic deviations are associated with neuromuscular imbalance between the vastus lateralis (VL) and vastus medialis (VM), leading to abnormal patellar tracking, increased patellofemoral joint stress [11,12,13], and weakness of the hip abductors and external rotators [14,15,16]. Reduced hip abductor strength increases the hip adduction angle during running, whereas a quadriceps activation imbalance contributes to greater knee flexion angles, compromising patellofemoral joint stability [12,14,16]. Additionally, individuals with PFP often present with delayed gluteus medius (GMed) activation and increased iliotibial band (ITB) stiffness [17,18,19].

Decreased ITB flexibility may restrict hip and knee mobility and negatively affect neuromuscular control during running [20]. Furthermore, the hip adduction angle during running correlates poorly with GMed activation, but positively with late activation of the tensor fascia late (TFL) [21]. The TFL is linked to the ITB and functions with the gluteal muscles to store and release elastic energy while running, thereby contributing to pelvic stability [22,23]. This stability is reduced in individuals with PFP due to various dysfunctions, including the frequent presence of latent myofascial trigger points (MTrPs) in GMed and peripatellar regions [24,25]. Therefore, treatment with GMed MTrPs may play a role in decreasing the symptoms of PFP.

Instrument-assisted soft tissue mobilization (IASTM) is a therapeutic technique that employs stainless steel tools to enhance connective tissue mobility [26]. Originating from the traditional Asian practice of Gua Sha, IASTM has been modernized and adapted for clinical use [27]. These tools are applied to the skin, muscles, fascia, and tendons to break down adhesions and stimulate localized blood flow, thereby promoting tissue healing, reducing pain, and improving range of motion (ROM) and muscular strength [26,28,29,30]. IASTM significantly increases pain pressure thresholds when applied to trigger points [31] and can improve hip flexion/extension strength and ROM [32,33,34,35]. Also, the technique improves ROM more rapidly than stretching [36]. Furthermore, Hains et al. [24] suggest that PFP can be effectively managed using myofascial release techniques.

Recent studies on IASTM for PFP have demonstrated that IASTM may be more effective than massage or stretching techniques, leading to more sustained pain reduction and better improvements in strength and function [29,37]. However, these studies combined IASTM with exercise interventions or employed comparative designs between different manual therapy approaches [29,37]. To date, the examination of the effects of IASTM as a standalone intervention is limited. There is a lack of research that varies treatment sites base on the underlying biomechanical issues of PFP (e.g., hip muscle weakness and MTrP in female runners).

PFP in female runners is often associated with quadriceps muscle imbalance and hip abductor weakness [12,14]. IASTM has been proposed as a therapeutic approach to address these impairments, and its application to both the hip and knee may provide superior clinical outcomes compared to knee-only treatment in female runners. Therefore, this study aimed to compare the immediate and short-term (1-week) effects of a hip and knee IASTM protocol a longer-duration intervention versus a knee-only IASTM protocol a short-duration intervention on pain and function in female runners with PFP. We hypothesized that adding hip treatment would result in greater reduction in running related pain. Secondary hypothesis was that hip and knee treatment would lead to improvements in hip adduction ROM, hip muscle strength, and functional performance.

## 2. Materials and Methods

### 2.1. Participants

Twenty-eight female runners with PFP were recruited via social media from running clubs located in Seoul and Gyeonggi. Participants were recreational runners aged between 18 and 45 years who ran at least three times per week, with an average of more than 6 km per session [18,38,39,40]. Participants were required to have an insidious onset of anterior or peripatellar knee pain during running without any history of direct trauma; pain intensity during running had to be ≥3 on a visual analog scale (VAS). Additionally, the participants needed to have pain during at least one activity involving patellofemoral loading such as squatting, stair ascent/descent, jumping, or prolonged sitting [41]. The duration of symptoms had to exceed 2 months, and palpation of the medial or lateral patellar facets had to reproduce the pain [41,42]. All participants had a hip adduction angle of <24° on Ober’s test based on normative values (mean, 23.16°) established in a previous study evaluating ITB flexibility [43]. We excluded participants with a history of lower limb surgery within the past 12 months or the presence of any of the following conditions: (1) ligamentous laxity; (2) meniscal pathology; (3) synovitis; (4) patellar dislocation or subluxation; (5) patellar tendinopathy; (6) neurological impairments following spinal surgery; (7) use of pain relieving medications (e.g., corticosteroids or NSAIDs); (8) Osgood–Schlatter or Sinding–Larsen–Johansson syndrome; (9) foot deformities such as flat feet; (10) pregnancy; and (11) aggravation of symptoms in the foot, ankle, hip, or lower back during running [41,44,45]. For bilateral knee pain, the limb with more severe symptoms was selected for the intervention. All participants provided informed consent and were randomly assigned to the hip and knee (HK) (n = 14) or knee (K) (n = 14) groups.

### 2.2. Sample Size Calculation

This study was approved by the Institutional Review Board of CHA University (No. 1044308-202502-HR-232-02) and conducted in accordance with the ethical principles of the Declaration of Helsinki. The sample size was calculated using G*Power software (version 3.1.9.7), based on a repeated measures analysis of variance (ANOVA) (within between interaction), with an effect size (f) of 0.25, α level of 0.05, 1- β err prob of 0.80, two groups, and three measurements. These parameters were adopted from a previous study comparing the effectiveness of soft tissue techniques and massage in individuals with PFP [30]. The assumed correlation among repeated measures was set at 0.5, with a non-sphericity correction of 1.0. The analysis indicated that 28 participants (14 in each group) were required to achieve sufficient statistical power.

### 2.3. Study Protocol

All measurements and treatment procedures were performed at the Sports Medicine Graduate Laboratory of Athletic Training at the CHA University, Gyeonggido, Republic of Korea. A total of 28 participants who met the study criteria were randomly assigned to either the HK or K group, using a computer-generated random sequence. Allocation was concealed by sequentially numbering and sealing the envelopes. All interventions were delivered by a researcher, while all data collection and measurements were performed by a separate research assistant with over 10 years of experience who was blinded to group allocation. Demographic data, including height, weight, body mass index (BMI), age, and weekly running distance, were recorded. Pain experienced during running over the past week was assessed using a VAS, followed by measurements of hip adduction ROM, hip abduction strength, and hip external rotation strength. Functional capacity was then assessed using the step-down test, with performance quantified by the number of repetitions, and pain intensity was measured using the VAS.

The participants then underwent IASTM. Immediately after treatment, hip adduction ROM, hip abduction strength, hip external rotation strength, and step-down tests were performed. All measurements were repeated 1 week later to identify the effects. During the 1-week period, participants were instructed to continue running the program without any new training or treatment (Figure 1).

### 2.4. Outcome Measures

#### 2.4.1. Visual Analog Scale—Worst Pain (VAS-W)

The VAS was used to evaluate the intensity of the PFP before and after the intervention. A 10-cm line was employed, with 0 cm indicating no pain and 10 cm representing the worst imaginable pain. The participants were instructed to mark the point on the scale that corresponded to the worst knee pain they experienced during running in the past week, as well as the worst pain felt during the step-down test. Pain during the step-down test was measured immediately after the intervention and again 1 week later. Pain during running was reassessed at the 1-week follow-up [46,47].

#### 2.4.2. Hip Adduction ROM

Hip adduction ROM was measured using Ober’s test with the Goniometer Pro application on an iPhone 12 (Apple Inc., Cupertino, CA, USA) (Figure 2) [48,49]. The participants lay on their side with the non-tested limb flexed at the hip at 45° and the knee at 90°, while the tested limb was placed on top. The examiner positioned the iPhone 12 over the lateral epicondyle of the femur. With the knee of the tested limb flexed to 90°, the examiner passively abducted and extended the hip until the trunk and legs were aligned. To prevent internal femoral rotation and hip flexion, the examiner stabilized the medial side of the knee and gradually lowered the limb. The point at which resistance was experienced or pelvic rotation initiated was recorded as the end ROM. Values below the horizontal (indicating adduction) were recorded as positive, whereas values above the horizontal (indicating abduction) were recorded as negative [19]. Measurements were performed twice and the mean value was used for analysis [48,50]. The intraclass correlation coefficients (ICC) ranged from 0.95 to 1.0 [51]. Based on a previous study that identified the normative cutoff value for ITB flexibility as 23.16°, participants with hip adduction ROM greater than 24° were considered to have no ITB tightness and were excluded from the study [43].

#### 2.4.3. Hip Strength

Hip muscle strength was assessed using a Lafayette 01165A handheld dynamometer (Lafayette Instrument, Lafayette, IN, USA) equipped with a contoured foam-padded stirrup. Measurements were performed at three points: pre-intervention, immediately post-intervention, and at the 1-week follow-up. During testing, the examiner provided consistent resistance, and participants were instructed to exert maximal force against the resistance. The measured force values were normalized to body weight using the following formula: (kg force/kg body mass) × 100 [50].

#### 2.4.4. Hip Abduction

The participants were placed in a side-lying position with the non-tested limb flexed at the hip to 45° and the knee to 90°. The tested limb was placed in 30° hip abduction, 5° hip extension, and neutral rotation. The examiner stabilized the pelvis with one hand and applied resistance proximal to the lateral malleolus with the other hand [52]. Participants were instructed to gradually increase force for over 5 s. Following a “ready–set–go” cue, two trials of maximum isometric contraction were performed after one practice attempt, with a 1-min rest interval between trials. The average of the two maximal efforts was recorded (Figure 3A) [50]. Visual monitoring was conducted throughout the test to ensure that the hip remained in the proper position and that no flexion occurred. The intra-rater reliability for this test ranges from 0.93 to 0.94 [52].

#### 2.4.5. Hip External Rotation

The participants lay prone on the examination table with the hip in a neutral rotational position and the knee flexed at 90°. Resistance was applied proximal to the medial malleolus [52], and the examiner stabilized the pelvis with one hand to prevent compensatory trunk or pelvic rotation. As with the hip abduction test, one practice trial was followed by two trials of maximum isometric contraction. The average of two measurements was recorded for analysis (Figure 3B) [50]. The intra-rater reliability for this test has been reported with an ICC of 0.93–0.97 [52].

### 2.5. Functional Performance and Pain

#### Step-Down Test

To evaluate functional performance and patellofemoral joint-related pain, a step-down test was performed at three time points: pre-intervention, immediately post-intervention, and at a 1-week follow-up (Figure 4) [53]. The participants stood on an 8-inch (20 cm) box with the tested limb, while the non-tested limb was extended forward. The participants were instructed to gently lower the heel of the non-tested limb to touch the ground without transferring weight and then return to the starting position. The participants repeated this movement for 30 s, and the total number of completed repetitions was recorded. The pain experienced during the task was assessed using a VAS [47,54].

### 2.6. IASTM Application

All IASTM procedures were performed by a certified athletic trainer with 3 years of clinical experience. The HK group received a single 7-min session of IASTM targeting the quadriceps, VM, VL trigger points, patellar borders, ITB, ITB–TFL interface, and ITB–GMed interface. The K group received a 3-min IASTM session targeting the quadriceps, VM, and VL trigger points and patellar borders (Figure 5). IASTM was applied after the pre-intervention assessments and a 5-min treadmill warm-up at a walking speed of 6 km/h [55].

Bow and Arrow Soft Tissue Mobilization (BASTM, Seoul, Republic of Korea) was used for the IASTM procedures (Figure 6). A lubricant was applied over the treatment area to reduce skin friction. The instruments were held at a 45° angle to the skin surface for treatment. The bow was used to scrape the superficial soft tissues, and the arrow was used to release the scar tissue and adhesions.

Participants were seated with their knees flexed at 90°. Using the convex side of the bow, a stroke was applied along the anterior surface of the thigh from the base of the patella to the anterior inferior iliac spine for 30 s. The head of the arrow was scraped for 30 s in the direction of the muscle fibers of the entire quadriceps group, including the VM, vastus intermedius (VI), rectus femoris, and VL (Figure 7B) [32,33].

Participants were seated with the knee flexed at 90°, and the head of the arrow was used to stroke the mid belly of the VL and the trigger points within the vastus medialis oblique (VMO) for 1 min (Figure 8A,B) [56,57,58].

The head of the arrow was used for 30 s around the patellar borders, and the tail of the arrow was used to stroke for 30 s along the patella (Figure 9) [24].

#### 2.6.1. ITB

Participants were placed in a side-lying position. The convex bow was stroked for 1 min from the lateral joint line of the knee to the iliac crest (Figure 10 and Figure 11A). The head of the arrow was used to stroke specific regions for 1 min, including the area where Gerdy’s tubercle, the patellar retinaculum, and the VL trigger point overlap, as well as the region between the anterior superior iliac spine and the greater trochanter (Figure 12A) [37].

#### 2.6.2. ITB and TFL/GMed Border

In the side-lying position with both hips flexed at 45° and knees flexed at 90°, the TFL was scraped with the head of the arrow for 1 min, focusing on localized tender points. The treatment was applied from the distal fascial insertion between the gluteus maximus and TFL to the anterior superior iliac spine (Figure 12B) [24]. For the GMed, the area between the greater trochanter and the inferior border of the iliac crest in the upper lateral quadrant of the buttock was scraped with the head of the arrow for 1 min (Figure 12C) [25,58,59].

### 2.7. Statistical Analysis

All statistical analyses were performed using IBM SPSS Statistics (version 27.0; SPSS Inc., Armonk, NY, USA). Participant demographic characteristics (age, height, weight, BMI, and weekly running distance) are reported as means and standard deviations (mean ± SD). Baseline homogeneity of continuous demographic variables (age, height, weight, BMI, running distance) was assessed using independent sample *t*-tests. The normality of the data was tested using the Shapiro–Wilk test. Variables that met the assumption of normality were analyzed using two-way repeated-measures ANOVA. Running-related pain was analyzed using a 2 × 2 two-way repeated-measures ANOVA (group × time; *p* < 0.05). Hip adduction ROM, hip muscle strength, and step-down functional performance were analyzed using a 2 × 3 two-way repeated-measures ANOVA (group × time; *p* < 0.05). The step-down VAS scores did not meet the assumption of normality. Therefore, a rank-based two-way repeated-measures ANOVA (group* time) was used to analyze group-and- time interaction effect. Within-group difference across time points were analyzed using the Wilcoxon signed-rank test. When significant main or interaction effect were found in rank-based ANOVA, post-hoc comparisons with Bonferroni corrections were applied Effect sizes were reported using partial eta squared (η^2^) and r values. Effect sizes were interpreted as trivial (≤0.20), small (0.21–0.49), moderate (0.50–0.79), or large (≥0.80). All statistical tests were evaluated using a 95% confidence interval, and a *p*-value < 0.05 was considered statistically significant.

## 3. Results

The physical characteristics of the participants are presented in Table 1. No significant differences were observed between the groups in age (*p* = 0.722), height (*p* = 0.861), weight (*p* = 0.089), BMI (*p* = 0.073), and weekly running distance (*p* = 0.868). The pre-intervention, immediate post-intervention, and 1-week follow-up values for the primary outcome measures are presented in Table 2. Running-related pain showed a significant reduction from baseline (HK: 5.49 ± 2.14; K: 5.30 ± 1.45) to week1 (HK: 1.30 ± 1.08; K: 1.57 ± 1.20) (F (1,26) = 144.400, *p* < 0.001, η^2^ = 0.847). However, the mean reduction did not significantly differ between the groups (*p* = 0.494). Post-hoc analysis revealed a significant reduction in pain from baseline to 1 week after the intervention (*p* < 0.001) (Table 3). The results for all three points are listed in Table 4. Hip adduction ROM demonstrated a significant main effect of time (F (2,52) = 19.508, *p* < 0.001, η^2^ = 0.139), with no significant interaction between groups (F (2,52) = 0.519, *p* = 0.598, η^2^ = 0.020). Post-hoc comparisons revealed significant improvements from baseline to immediately after the intervention and from baseline to one week later (both *p* < 0.001). Hip abduction strength showed a significant main effect of time (F (2,52) = 4.199, *p* = 0.020, η^2^ = 0.139), while the group-by-time interaction was not statistically significant (F (2,52) = 2.992, *p* = 0.059, η^2^ = 0.103). Post-hoc analysis indicated a significant improvement from baseline to immediately after the intervention (*p* = 0.009). Hip external rotation strength did not show a significant main effect of time (F (2,52) = 0.188, *p* = 0.829, η^2^ = 0.007) or group interaction (F (2,52) = 0.702, *p* = 0.500, η^2^ = 0.026). Step-down functional performance showed a significant main effect of time (F (2,52) = 47.250, *p* < 0.001, η^2^ = 0.645), with no significant group-by-time interaction (F (2,52) = 0.431, *p* = 0.652, η^2^ = 0.016). Post-hoc comparisons revealed significant improvements from baseline to immediately after the intervention and 1 week later (both *p* < 0.001). Pain during the step-down test did not significantly differ between the groups at time point (*p* = 1.000) and no significant group-by-time interaction was observed (*p* = 0.109) (Table 5). However, post-hoc Wilcoxon signed-rank test indicated significant differences over time, with the largest effect size between baseline and week 1 (*p* < 0.001, r = 0.874), followed by baseline vs. immediate (*p* < 0.001, r = 0.855) and immediate vs. week 1 (*p* = 0.044, r = 0.381) (Table 6). This discrepancy may be attributed to differences in the analytical approach: the repeated measures ANOVA assesses overall trends across all time points and accounts for within-subject variance, whereas the Wilcoxon test evaluates pairwise differences between specific time points.

## 4. Discussion

We aimed to compare the immediate and short-term (1-week) effects of a hip-and-knee IASTM protocols a longer-duration intervention versus knee-only IASTM protocol a shorter-duration intervention on pain and function in female recreational runners with PFP. Following a single intervention session, both groups showed a significant reduction in pain during running and step-down tasks. Furthermore, a secondary outcome, Hip adduction ROM and functional performance also improved immediately after the intervention, and these effects were maintained at the 1-week follow-up. Our findings suggest that a single IASTM session provided short-term benefits in reducing pain and improving function in female runner with PFP, regardless of the treatment region and duration. However, the lack of significant differences in muscle strength outcomes warrants further investigation to determine whether additional hip-focused treatments can effectively enhance abduction strength and mitigate excessive hip adduction patterns.

A single IASTM session resulted in a significant reduction in running-related pain at the 1-week follow-up and a consistent reduction is step-down pain immediately after intervention until week 1later in both groups. However, no significant difference was observed between the two groups, regardless of the treatment site. This analytical approach objectives and data characteristics perception of their correction pain and contributed to the observed reduction immediately after intervention. In a study by Gulick [31], it was reported that a 5-min IASTM intervention can effectively reduce the pain pressure threshold of a MTrP over a three-week period. These findings are consistent with those of previous studies reporting that IASTM applied to various MTrP regions can effectively reduce pain and improve function [31,60,61]. Studies on PFP populations show that IASTM produces immediate pain relief that is sustained for up to 4 weeks [29,30], and weekly assessments of pain intensity have demonstrated significant reductions even after 1 week [37]. However, despite the within-group improvements, no significant differences were observed between our two treatment groups. Both groups underwent IASTM targeting anatomical areas commonly associated with high MTrP prevalence in individuals with PFP. MTrPs are classified as active or latent, with active MTrPs being spontaneously painful and easily recognized by patients [62]. Studies investigating the MTrP distribution in PFP populations have reported a high prevalence of latent MTrPs in the GMed and active MTrPs in the VL [25,58]. Given the high prevalence of active MTrPs in the quadriceps and around the patella, both treatment protocols in this study may have had a direct effect on participants’ pain perception, thus contributing to the similar levels of pain reduction observed in both groups. Future studies should consider the effects of longer treatment periods of IASTM in female runners with PFP.

In this study, both groups demonstrated significant improvements in hip adduction ROM immediately after the intervention, and these effects were sustained 1 week later (Table 4). Unuvar et al. [26], observed significant improvements in hip adduction ROM after a 6-week IASTM intervention targeting the thigh and peri-hip musculature. The study results are consistent with ours and support the effectiveness of a single IASTM session in improving hip adduction ROM. However, no significant differences were observed between groups. Hip adduction ROM was assessed using Ober’s test, which, according to Willett et al. [63], can be influenced not only by ITB stiffness but also by the surrounding soft tissues. However, no significant difference was observed between groups. Hip adduction ROM was assessed using Ober’s test, which, according to Willett et al. (2016), can be influenced not only by ITB stiffness but also by surrounding soft tissues [63]. Fairclough et al. noted that the distal ITB inserts into Gerdy’s tubercle and is closely interconnected with the VL when the knee is flexed at approximately 30° [64]. In individuals with PFP, a weakened VM combined with tension in the ITB and VL is associated with patellar maltracking [65]. Additionally, patients with PFP often demonstrate compensatory strategies involving increased activation of both VL and VM [66]. The IASTM of the VL and ITB may have influenced the lateral fascial system, contributing to the increased hip adduction ROM. Although this immediate benefit was observed in both groups, no additional effect was observed with hip or knee treatment. Therefore, future studies should investigate whether extended or repeated IASTM protocols targeting the hip region provide superior benefits compared with knee-only interventions.

Hip abduction strength showed a significant main effect of time, with an increase observed immediately after the intervention compared with baseline. However, this improvement was not maintained at the 1-week follow-up and there was no statistically significant difference between the groups (*p* = 0.059). Dierks et al. reported that individuals with PFP exhibit reduced hip abduction strength at both the beginning and end of a running session [16]. Since hip abduction strength contributes to pelvic stability and controls femoral motion, improvements in this capacity may be critical for managing excessive femoral movement in patients with PFP [15]. Studies have reported an increase in hip strength after the IASTM. Pisirici et al. observed enhanced hip abduction strength in individuals with dynamic knee valgus after IASTM treatment [67], and Unuvar et al. found similar outcomes in athletes with ITB tightness [26]. Moreover, studies involving patients with PFP reported gains in hip extension strength after IASTM treatment [29,30]. These findings support the results of the present study and suggest that improvements in muscle function and contractile strength may be mediated through enhanced blood flow and the removal of adhesions within myofascial tissues [28,35,68,69]. The lack of statistically significant group differences in the present study may be attributed to the single-session nature of the intervention, whereas previous studies often utilized repeated treatments over a 6-week period with a frequency of at least twice per week. Future research should include larger sample sizes and repeated treatment protocols to assess the potential benefits of hip-specific IASTM on strength gain more clearly.

Although strengthening the hip external rotators is essential in runners with PFP, as increased hip internal rotation during running can lead to abnormal femoral rotation [70], we found no significant change in hip external rotation strength over time or between the groups following IASTM treatment. This lack of change may be attributed to the anatomical focus of the IASTM intervention. The treatment primarily targeted the anterior fibers of the GMed, TFL, and their connections to the ITB. These muscles are primarily involved in hip abduction and internal rotation, particularly when the hip is flexed beyond 30° [71]. Moreover, true hip external rotation is governed by deeper musculature such as the obturator internus and quadratus femoris [71]. This lack of change may be attributed to the anatomical focus of the IASTM intervention. The treatment primarily targeted the anterior fibers of the GMed, the TFL, and their connections to the ITB. Previous studies have shown that these muscles are primarily involved in hip abduction and internal rotation, particularly when the hip is flexed beyond 30 degrees [71]. Moreover, true hip external rotation is predominantly governed by deeper musculature such as the obturator internus and quadratus femoris [71]. Given that the IASTM treatment in this study did not directly target the deeper external rotators, the intervention may have had a limited influence on external rotation strength. Future research should consider incorporating targeted IASTM protocols that address the deep external rotator muscles to assess their contribution to hip strength improvement in individuals with PFP more accurately.

Both groups showed immediate improvements in This immediate effect could have influenced participants’ step-down performance, and these effects were maintained at the 1-week follow-up. The step-down test is a reliable tool for assessing functional performance in individuals with PFP, and is particularly sensitive in female populations [47,53]. In the study by Glaviano et al., women with PFP demonstrated reduced activation of both the quadriceps and gluteal muscles during the step-down task, with relatively higher activation of the VMO compared to the VL [72]. Similarly, Elmahdy et al. reported delayed activation onset of the VL in runners with PFP, suggesting impaired neuromuscular control [45]. The baseline step-down performance of participants in the present study was lower than the average repetitions reported in previous studies (15.90 ± 5.4), indicating a greater initial functional limitation. Post-intervention performance improvements may be attributed to a number of factors. The observed gains could reflect neuromuscular adaptation, pain relief, or placebo response. Because of the study design, we were unable to directly assess the kinematic changes that would confirm this mechanistic relationship. While the HK group demonstrated a greater mean increase in repetitions than the K group, the between-group difference was not statistically significant. This may be due to the limited increase in hip abduction strength and the fact that step-down performance is more sensitive to quadriceps function than to hip abductor strength. Functional deficits during step-down tasks have been associated with weakened hip abduction strength, reduced quadriceps strength, and decreased flexibility of the ITB and TFL [73]. Previous studies have also reported impaired eccentric control of the quadriceps in individuals with PFP [74]. Although the hip adduction ROM improved in both groups in this study, the increase in hip abduction strength, which was not statistically significant (*p* = 0.059), may not have been sufficient to produce a significant improvement in functional performance. Future research should incorporate assessments of quadriceps flexibility, strength, and lower limb kinematics during step-down tasks to identify key contributors to functional improvement in individuals with PFP.

This study had some limitations. First, the runners had a relatively low weekly running volume. While previous studies defined recreational runners with PFP as those running >15 km/week [75], the participants in this study averaged approximately 6 km/week, indicating a lower training load. Therefore, future studies should include amateur runners with higher weekly mileage to enhance the generalizability of the findings. Second, this study examined only the immediate and 1-week short-term effects of a single IASTM treatment. The absence of statistically significant differences between the two treatment protocols may be attributed to the limited nature of a single session intervention. Long-term effects and between-group differences may become more apparent with repeated treatments and extended follow-up. Thus, future research should employ a larger sample size and longitudinal design to evaluate the sustained impact of repeated IASTM applications. Third, this study is the lack of a non-intervention control group. This makes it difficult to definitively determine whether the observed improvements in pain and function are attributable to the IASTM intervention or the natural course of symptom recovery. Future studies should include a control group to account for the natural progression of symptoms. Finally, this study was conducted exclusively on recreational female runners, which limits the applicability of the findings to broader populations. Further studies are warranted to determine whether similar effects can be observed in other demographic groups, including male runners, elite athletes, or individuals with different activity levels.

## 5. Conclusions

A single IASTM application was effective for immediate pain relief and functional improvement in recreational female runners with PFP for at least 1 week. Both the HK and K groups showed reductions in running-related and step-down pain immediately after the intervention, along with improvements in ROM and patellofemoral function. Immediate IASTM treatment can provide similar short-term effects regardless of the application site (knee alone or knee and hip) despite the biomechanical factors contributing to PFP in female runners. Therefore, future research should incorporate repeated hip-targeted interventions, varying IASTM intensities, and treatments tailored to the anatomical location of MTrPs to evaluate the long-term effects and strength-related outcomes more comprehensively.

## Figures and Tables

**Figure 1 medicina-61-01912-f001:**
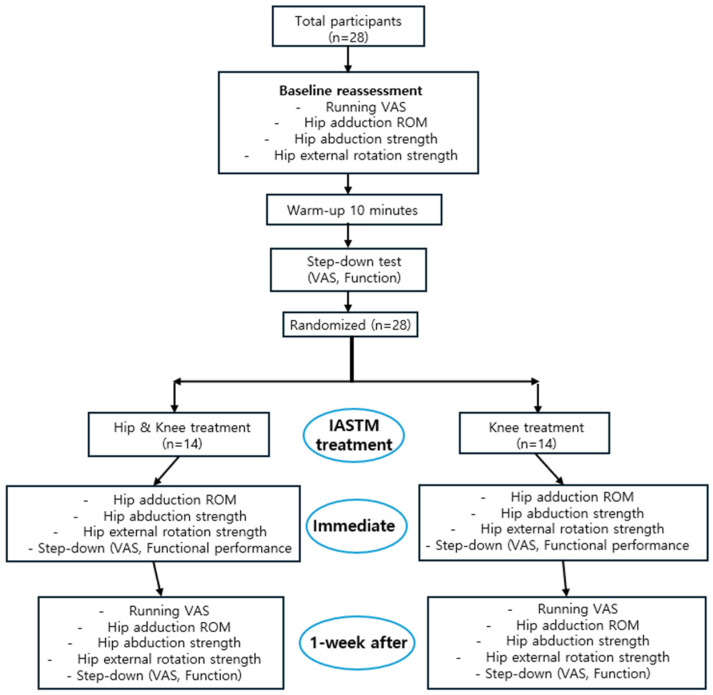
Flow chart of the protocol of the study.

**Figure 2 medicina-61-01912-f002:**
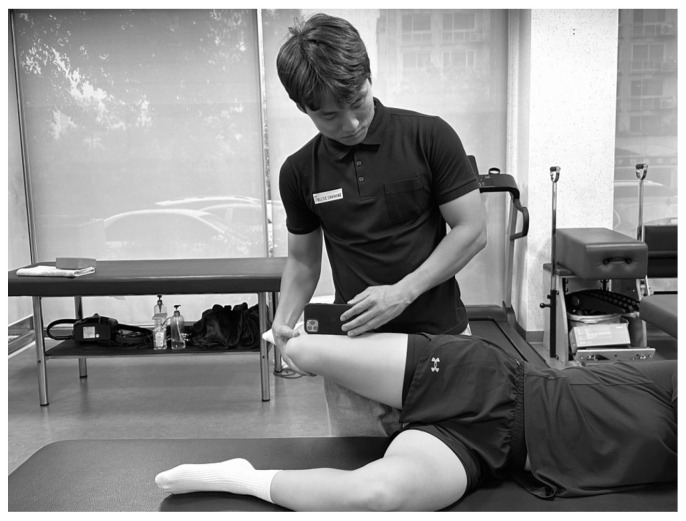
Ober’s test.

**Figure 3 medicina-61-01912-f003:**
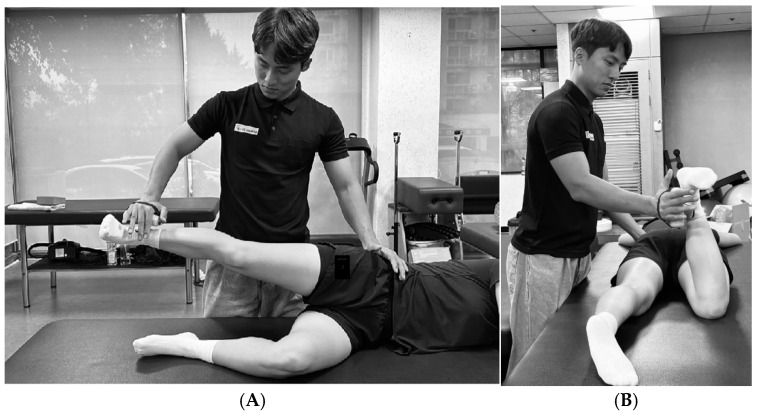
Strength measurements. (**A**). Measurement of hip abduction strength. (**B**). Measurement of hip external rotation strength.

**Figure 4 medicina-61-01912-f004:**
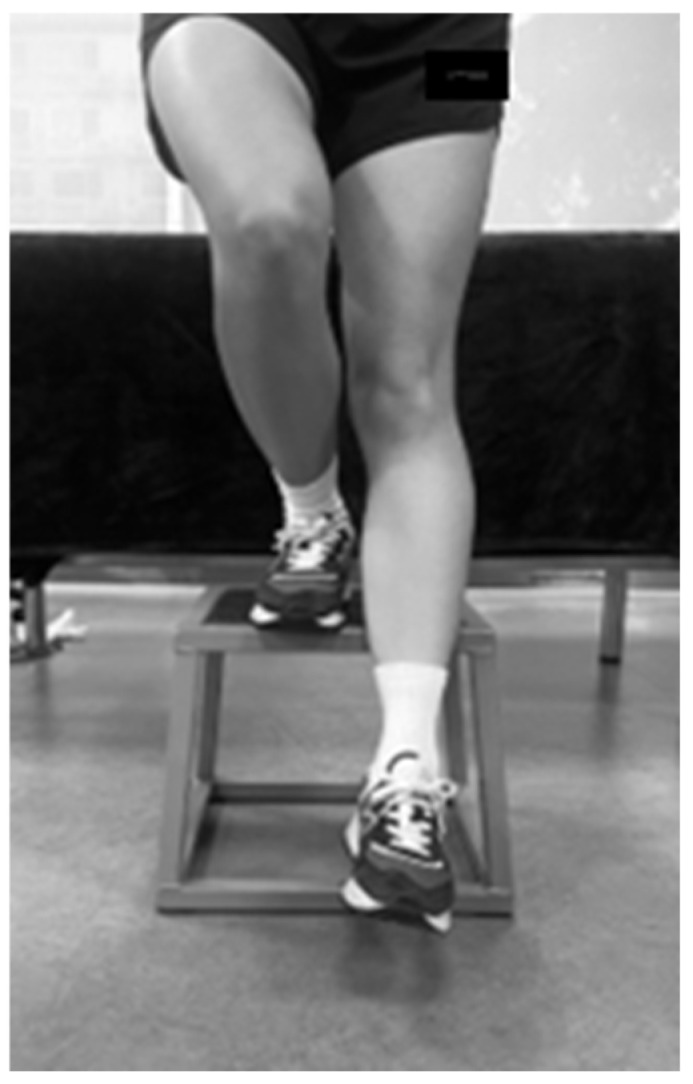
Step-down test.

**Figure 5 medicina-61-01912-f005:**
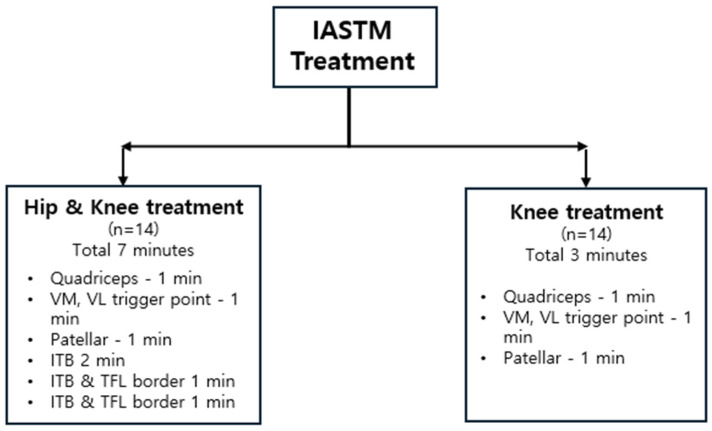
IASTM protocol.

**Figure 6 medicina-61-01912-f006:**
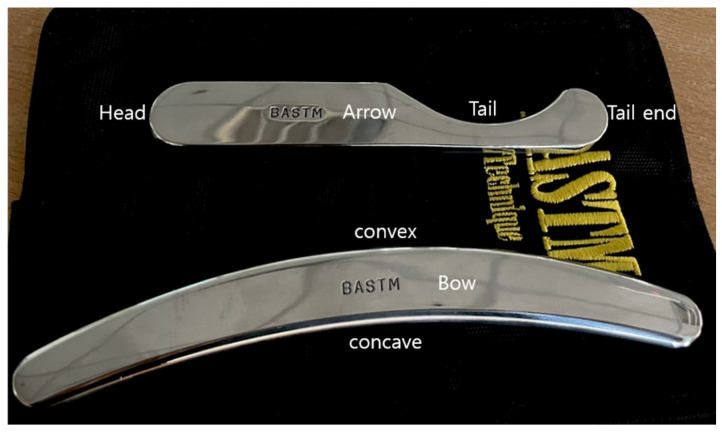
Bow and Arrow Soft Tissue Mobilization (BASTM) tool.

**Figure 7 medicina-61-01912-f007:**
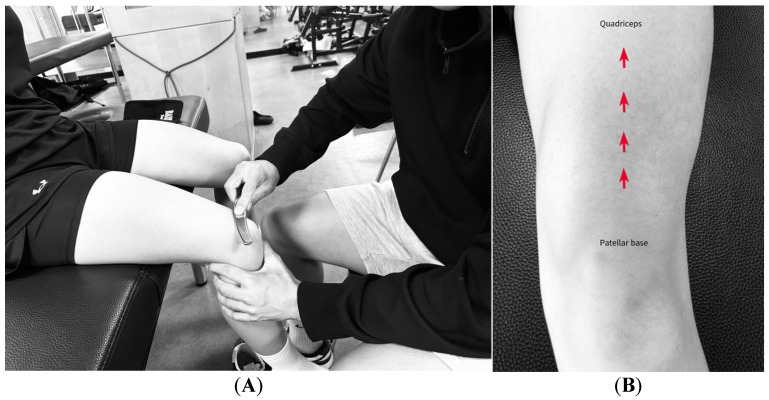
Implementation of BASTM on the quadriceps. (**A**) Application of BASTM to the quadriceps; (**B**) Direction of BASTM stroke on the quadriceps muscle group.

**Figure 8 medicina-61-01912-f008:**
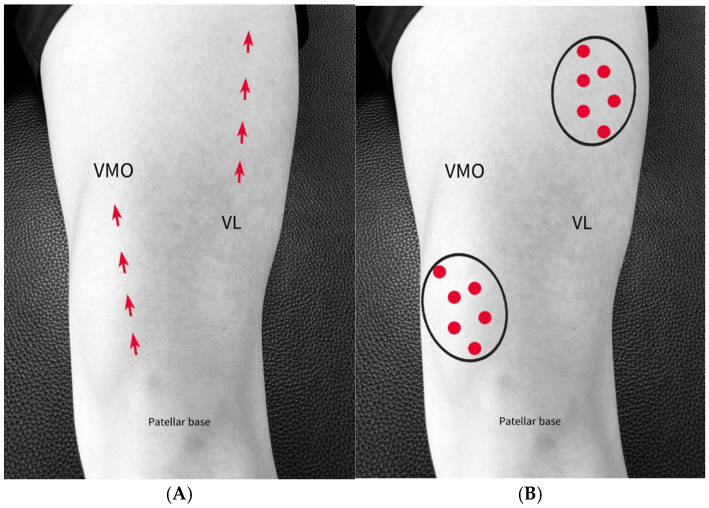
Implementation of BASTM on VM and VL. (**A**) Instrument stroke direction on the VMO and VL; (**B**) Location of active and latent MTrPs in the VM and VL.

**Figure 9 medicina-61-01912-f009:**
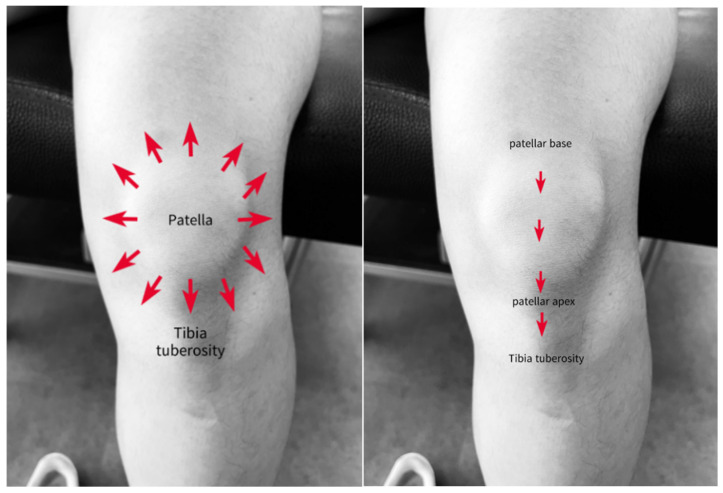
Implementation of BASTM on the patella.

**Figure 10 medicina-61-01912-f010:**
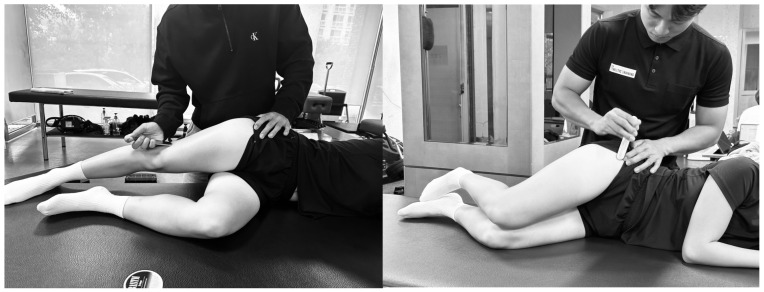
Implementation of BASTM on the ITB.

**Figure 11 medicina-61-01912-f011:**
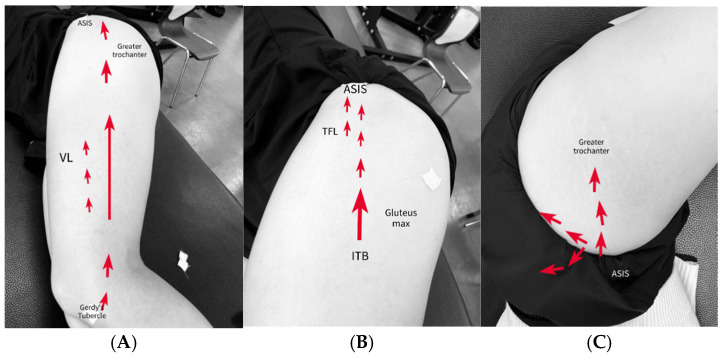
Implementation BASTM on the ITB and TFL/GM border. (**A**). Stroking on the ITB. (**B**). Stroking the ITB border. (**C**). Stroking the Gmed/ITB border.

**Figure 12 medicina-61-01912-f012:**
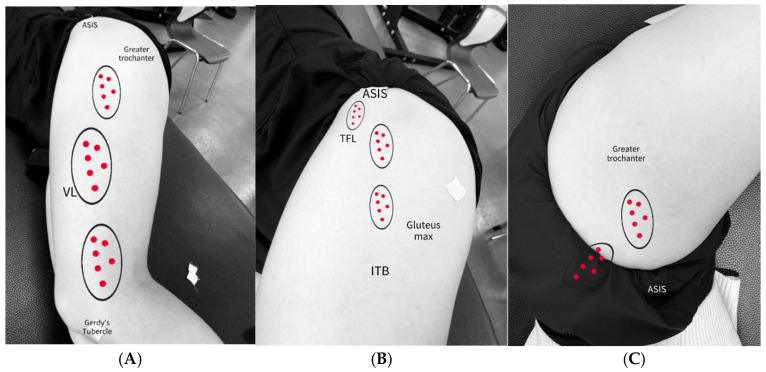
Distribution of MTrPs in the ITB, including TFL, and Gmed. (**A**). Overlapping regions between the VL and ITB, and in the ITB alone; (**B**). ITB and TFL border; (**C**). ITB and Gmed border.

**Table 1 medicina-61-01912-t001:** Demographic characteristics of the participants.

	HK (n = 14)	K (n = 14)	t	*p*
Age (years)	23.39 ± 2.55	25.75 ± 4.16	0.360	0.722
Height (m)	1.62 ± 0.05	165.67 ± 4.66	0.176	0.861
Weight (kg)	55.9 ± 5.19	58.50 ± 5.68	1.764	0.089
BMI	21.4 ± 1.43	20.4 ± 1.43	1.869	0.073
Running distance (km) (week)	16.6 ± 6.62	16.9 ± 7.73	−0.131	0.868

Values are presented as mean ± standard deviation or n (%). Age, height, weight, and BMI were analyzed using an independent *t*-test, while running distance was analyzed using the Mann–Whitney U test. HK, hip and knee; K, knee; t, independent *t*-test statistic.

**Table 2 medicina-61-01912-t002:** Mean and standard deviation for data point.

Variable	Group	Baseline (95% CI)	Immediate (95% CI)	Week 1 (95% CI)
Running VAS	HK	5.49 ± 2.14 (4.28–6.68)	-	1.30 ± 1.08 * (0.69–1.90)
	K	5.30 ± 1.45 (4.46–6.13)	-	1.57 ± 1.20 * (0.87–2.26)
Hip AD ROM	HK	11.28 ± 3.91 (9.01–13.53)	15.41 ± 4.06 * (13.07–17.75)	14.37 ± 5.04 *^†^ (11.45–17.26)
	K	11.10 ± 3.51 (9.07–13.12)	14.51 ± 4.44 * (11.94–17.08)	14.83 ± 5.01 *^†^ (11.93–17.71)
Hip AB Strength	HK	14.11 ± 2.34 (12.75–15.46)	15.45 ± 1.87 * (14.37–16.53)	15.41 ± 2.39 *^†^ (14.02–16.78)
	K	15.79 ± 3.02 (14.04–17.53)	16.01 ± 2.94 * (14.30–17.70)	15.79 ± 3.43 *^†^ (13.81–17.77)
Hip ER Strength	HK	17.91 ± 3.24 (16.04–19.78)	18.43 ± 2.55 (16.95–19.89)	17.44 ± 5.3 (14.35–20.53)
	K	18.03 ± 2.02 (16.85–19.19)	18.11 ± 1.96 (16.98–19.24)	18.47 ± 2.52 (17.01–19.92)
Step-down (number)	HK	8.79 ± 4.41 (6.24–11.32)	11.57 ± 4.13 * (9.18–13.95)	12.36 ± 4.67 *^†^ (9.66–15.05)
	K	11.00 ± 2.88 (9.33–12.66)	13.36 ± 3.67 * (11.23–15.47)	13.93 ± 4.68 *^†^ (11.22–16.63)
Step-down VAS	HK	3.44 ± 1.29 (2.69–4.18)	0.51 ± 0.66 * (0.12–0.88)	0.56 ± 1.02 *^†^ (0–1.15)
	K	3.26 ± 1.41 (2.45–4.07)	1.44 ± 1.39 * (0.64–2.24)	0.47 ± 0.59 *^†^ (0.23–1.15)

Values are presented as mean ± standard deviation or n (%) with 95% confidence intervals (CIs). The VAS scores ranged from 0 to 10, and lower CI bounds of < 0. VAS, Visual analog scale; ROM, Range of motion; AD, Hip adduction; HK, Hip and knee; K, Knee; AB, Hip abduction; ER, Hip external rotation. * *p* < 0.05 compared with baseline; ^†^ *p* < 0.05 compared with immediate. No significant differences were observed between the groups at any time.

**Table 3 medicina-61-01912-t003:** Two-way repeated-measures ANOVA results for running VAS (baseline vs. week 1).

Effect	df	F	*p*	η^2^
Time	1, 26	144.400	**<0.001 ****	0.847
Time * Group	1, 26	0.482	0.494	0.018

VAS, Visual analog scale; df, Degree of freedom; F, f-value; *p*, *p*-value; n^2^, partial n^2^; B, baseline; W, week 1; ** *p* < 0.001.

**Table 4 medicina-61-01912-t004:** Two-way repeated-measures ANOVA results for outcomes across three time points.

Variable	Effect	df	F	*p*	η^2^	Post-Hoc (Bonferroni)
Hip AD ROM	Time	2, 52	19.508	**<0.001 ****	0.139	B vs. I (*p* **< 0.001 ****)
	Time * Group	2, 52	0.519	0.598	0.020	B vs. W (*p* **< 0.001 ****)
						I vs. W (*p* = 1.000)
Hip AB Strength	Time	2, 52	4.199	**0.020 ***	0.139	B vs. I (*p* =.009 *)
	Time * Group	2, 52	2.992	0.059	0.103	B vs. W (*p* = 0.113)
						I vs. W (*p* = 1.000)
Hip ER Strength	Time	2, 52	0.188	0.829	0.007	B vs. I (*p* = 1.000)
	Time * Group	2, 52	0.702	0.500	0.026	B vs. W (*p* = 1.000)
						I vs. W (*p* = 1.000)
Step-down (number)	Time	2, 52	47.250	**<0.001 ****	0.645	B vs. I (*p* **< 0.001 ****) B vs. W (*p* **< 0.001 ****)
	Time * Group	2, 52	0.431	0.652	0.016	I vs. W (*p* = 0.218)

df, Degree of freedom; F, f-value; *p*, *p*-value; n^2^, partial n^2^; B, baseline; I, immediate; W, week 1; Hip AD ROM, Hip adduction range of motion; Hip AB, hip abduction; Hip ER, Hip external rotation. * *p* < 0.05, ** *p* < 0.001.

**Table 5 medicina-61-01912-t005:** Results the rank-based two way repeated-measures ANOVA for step-down VAS.

Effect	Values	F	df	df for Error	*p*
Time	0.000	0.000	2.000	25.000	1.000
Time * Group	0.163	2.430	2.000	25.000	0.109

df, Degrees of freedom; F, f-value; df, Degree of freedom; *p*, *p*-value.

**Table 6 medicina-61-01912-t006:** Wilcoxon signed-rank test results for within-group comparisons of step-down VASs.

Comparison	Z	*p*	Effect Size (r)
Baseline—Immediate	−4.524	<0.001 **	0.855
Baseline—Week1	−4.624	<0.001 **	0.874
Immediate—Week1	−2.015	0.044 *	0.381

Z, z-value; *p*, *p*-value; VAS, Visual analog scale. * *p* < 0.05, ** *p* < 0.001. Effect size, trivial (r ≤ 0.20), small (0.21 ≤ r ≤ 0.49), moderate (0.50 ≤ r ≤ 0.79), large (r ≥ 0.80).

## Data Availability

The original contributions presented in this study are included in the article. Further inquiries can be directed to the corresponding author.

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
