# Peer review of "The Immediate Effects of Instrument-Assisted Soft Tissue Mobilization on Pain and Function in Female Runners with Patellofemoral Pain"

_medicina, 2025, doi:10.3390/medicina61111912_

Round 1
Reviewer 1 Report
Comments and Suggestions for Authors
Abstract
The abstract overstates equivalence: it claims a single IASTM session is effective “regardless of whether the hip was included” and that effects are “similar” across sites, yet no between-group effect sizes or confidence intervals are reported—only p-values—so absence of a difference ≠ evidence of equivalence; the conclusion should be toned down or supported with between-group Δ and 95% CIs for a prespecified primary endpoint.
There’s no primary outcome named, and multiple endpoints are listed, inviting multiplicity and “cherry-picking” concerns; declare the primary endpoint and report its between-group result with a CI, moving the rest to key secondaries.
1. Introduction
The section reads like a mini-review but never distills to one falsifiable gap; the leap from prevalence/risk factors to IASTM is abrupt and under-justified. Epidemiology is mixed from non-comparable samples, which inflates the problem rather than sharpening the question. The biomechanics narrative asserts hip-driven valgus while acknowledging equivocal associations, yet uses that to justify adding hip treatment—this is a logical gap. Mechanistic rationale for standalone IASTM in PFP is thin (transient hypoalgesia/placebo not addressed), so there’s no causal bridge to the claimed functional effects. Prior evidence is listed, not synthesized to a decision point (superiority vs non-inferiority, MCID), and no single primary outcome is pre-declared.
The planned contrast is also confounded by dose (hip+knee gets more minutes than knee-only), and this isn’t acknowledged. Rewrite to: state one sharp research question, justify why IASTM should help PFP mechanistically, reconcile the hip-mechanics story, name a primary endpoint and comparative frame, and pre-empt the dose confound.
2. Methods
Allocation by “opaque envelopes” is under-described—no sequence generation, numbering, sealing, or role separation—so selection bias can’t be ruled out.
There’s no sham and no participant/assessor blinding; subjective pain and handheld strength tests result in substantial expectancy and measurement bias. You don’t declare a primary endpoint, yet test many outcomes across time without hierarchy or adjustment—this inflates Type I error and blurs inference.
The hip+knee arm receives ~7 minutes versus ~3 minutes in the knee-only arm, confounded by site and dose; either equalize exposure or frame this as a pragmatic unequal-dose comparison. The statistical plan leans on Kruskal–Wallis “across time,” which can’t test a group×time interaction; use a mixed-effects repeated-measures model (rank-based if needed).
Finally, intervention fidelity/measurement control are loose (no pressure/force targets, stroke counts, dynamometer fixation; single unblinded examiner).
3. Results
You report a lot of p-values but almost no between-group effect sizes or confidence intervals. With no prespecified primary endpoint, the section reads like a fishing expedition across many outcomes/timepoints. Pick the primary outcome and lead with the between-group Δ and 95% CI; demote the rest to key secondaries. As written, the “no interaction” findings can’t be read as “similar effects.”
There are internal inconsistencies that undermine confidence. In the text, hip external-rotation strength has a p-value of 0.737, while Table 4 shows a p-value of 0.829 for the time effect. Likewise, the hip adduction ROM time effect is reported as η² = 0.429 in the text but as η² = 0.139 in Table 4. These must agree.
You repeatedly lean on “trend toward significance” (p = 0.059 for hip abduction strength interaction). Either it meets your α or it doesn’t—report the point estimates with CIs and let readers judge precision; don’t imply a near-miss is meaningful.
Key clinical interpretability is missing. Nowhere do you state whether changes clear a known MCID or how many participants were “responders.” Give absolute changes (with 95% CIs) and MCID attainment rates for pain and function, not just significance labels.
4. Discussion
You repeatedly imply equivalence of sites (“similar short-term effects regardless of knee-only vs hip+knee”) without the design or reporting to support it. There is no sham/attention control; you did not prespecify a primary endpoint, and you report no between-group effect sizes with 95% CIs—so an interaction p > 0.05 cannot be interpreted as “same effect.” Either present prespecified non-inferiority margins (with Δ and CI) or temper the language to “no clear between-group differences were detected.
Your main contrast is also confounded by dose: hip+knee received roughly twice the IASTM time as knee-only. The Discussion treats “site” as the active ingredient while ignoring exposure time. You need to acknowledge that any apparent similarity (or trend) could reflect insufficient additional dose rather than true site-independence, and you should explicitly reframe claims as exploratory.
Mechanistic claims overreach the data. You ascribe changes in step-down performance and hip ROM to specific fascial/myofascial effects of IASTM, but with single-session, unblinded, short follow-up, those improvements are equally compatible with expectancy/hypoalgesia or generalized warm-up effects. Acknowledge this plainly and avoid causal narratives that you didn’t test (i.e., no kinematics, no blinded assessors, no fidelity/force control).
Comments on the Quality of English LanguageThe manuscript reads clearly enough to follow, but the prose could be tightened to improve first-pass comprehension. Several long sentences—especially in the Introduction—would benefit from being split, and the tense sometimes drifts (keep facts in present; Methods/Results in past). Terminology should be uniform once defined (e.g., use one label for the intervention throughout), and small style details need smoothing: consistent units and decimal places in tables, and a single format for statistics (p-values and 95% CIs). Figure captions can do a bit more work so a reader understands what was measured and how without returning to Methods. With these edits, the paper will read more naturally and present the science with less friction.
Reviewer 2 Report
Comments and Suggestions for Authors
The article titled “The Immediate Effects of Instrument-Assisted Soft Tissue Mobilization on Pain and Function in Female Runners with Patellofemoral Pain” presents a case-control study designed to evaluate the acute impact of instrument-assisted soft tissue mobilization (IASTM) on knee and hip function in female runners diagnosed with patellofemoral pain syndrome.
Major comments
The authors begin the introduction by stating that running is a popular physical activity with a high risk of overuse injuries, citing only one reference to support this claim. However, it is well established in the literature that running is associated with relatively low rates of severe injuries compared to other sports. It is essential to provide proper context for this information so that readers clearly understand the types of injuries most linked to running. Rather than focusing solely on percentages, as the authors do in subsequent sentences, it would be more informative to address the typical severity of these injuries.
The authors state that the sample size was calculated using G*Power software (version 3.1.9.7), based on a repeated-measures analysis of variance (ANOVA) (within- between interaction), with an effect size (f) of 0.25, an α level of 0.05, two groups, and three measurements. However, when replicating this information in the same software, the indicated N is 40 and not 28. Please explain the sample calculation process.
The authors state that Baseline homogeneity between groups was assessed using independent sample t-tests. How is this done? Wouldn't it be more appropriate to use a specific homogeneity test?
The authors also state that the normality of the data was tested using the Shapiro–Wilk test. Variables that met the assumption of normality were analyzed using two-way repeated-measures ANOVA. Were all the data normal? How were data that did not have normal distributions assessed?
It is uncommon for data derived from clinical interventions to follow a normal distribution; rather, it is much more typical to observe alternative distributions, such as logarithmic or Gamma distributions. Therefore, it is essential that the type of data distribution be taken into account. In this context, although ANOVA is a widely used statistical test, it does not appear to be the most appropriate statistical modeling approach for this type of study. I suggest that the authors consider employing Generalized Mixed Models (GMM or GEE) for the statistical analysis in the present study. It is highly likely that, due to the sample size being smaller than required, the authors attempted to evaluate the effect size to justify certain statistical recommendations. In this context, it would also be important to clarify the statistical power for each analysis performed. However, when employing models tailored to specific data distributions, such as Generalized Mixed Models (GMM or GEE), these parameters would not need to be assessed to justify the data.
Please explain how the randomization process was conducted.
The authors state in the discussion that, although no statistically significant differences were found between groups, a trend toward increased hip abduction strength was observed in the HK group. It is important to clarify what the authors define as a “trend” in this context.
Additionally, the authors should further discuss the results related to pain reduction. Was the reduction in pain potentially influenced by the application of the technique itself? The technique inherently involves applying pressure to the soft tissues, which may initially increase pain during its execution, yet subsequently result in immediate pain relief. Could this immediate effect obscure the participant’s perception of pain from the original injury?
The authors mention certain limitations; however, at no point do they address the possibility that the absence of a control group may be a significant limitation. It is essential to monitor the natural progression of pain symptoms to ensure that observed improvements are not merely a result of the passage of time or solely attributable to the intervention technique.
Minors’ Comments
It is necessary to make formatting adjustments throughout the entire text.
How were the demographic data, including height, weight, Body Mass Index (BMI), age, and weekly running distance, recorded?
